# Biomaterial-Assisted Anastomotic Healing: Serosal Adhesion of Pectin Films

**DOI:** 10.3390/polym13162811

**Published:** 2021-08-21

**Authors:** Yifan Zheng, Aidan F. Pierce, Willi L. Wagner, Hassan A. Khalil, Zi Chen, Charlotta Funaya, Maximilian Ackermann, Steven J. Mentzer

**Affiliations:** 1Laboratory of Adaptive and Regenerative Biology, Brigham & Women’s Hospital, Harvard Medical School, Boston, MA 02115, USA; yzheng@bwh.harvard.edu (Y.Z.); afpierce@bwh.harvard.edu (A.F.P.); willi.wagner@uni-heidelberg.de (W.L.W.); hakhalil@bwh.harvard.edu (H.A.K.); zchen33@bwh.harvard.edu (Z.C.); 2Department of Diagnostic and Interventional Radiology, Translational Lung Research Center, University of Heidelberg, 69117 Heidelberg, Germany; 3Electron Microscopy Core Facility, University of Heidelberg, 69117 Heidelberg, Germany; funaya@uni-heidelberg.de; 4Institute of Functional and Clinical Anatomy, University Medical Center of the Johannes Gutenberg-University, 55122 Mainz, Germany; maximilian.ackermann@uni-mainz.de

**Keywords:** bowel, serosa, biopolymer, pectin, heteropolysaccharide

## Abstract

Anastomotic leakage is a frequent complication of intestinal surgery and a major source of surgical morbidity. The timing of anastomotic failures suggests that leaks are the result of inadequate mechanical support during the vulnerable phase of wound healing. To identify a biomaterial with physical and mechanical properties appropriate for assisted anastomotic healing, we studied the adhesive properties of the plant-derived structural heteropolysaccharide called pectin. Specifically, we examined high methoxyl citrus pectin films at water contents between 17–24% for their adhesivity to ex vivo porcine small bowel serosa. In assays of tensile adhesion strength, pectin demonstrated significantly greater adhesivity to the serosa than either nanocellulose fiber (NCF) films or pressure sensitive adhesives (PSA) (*p* < 0.001). Similarly, in assays of shear resistance, pectin demonstrated significantly greater adhesivity to the serosa than either NCF films or PSA (*p* < 0.001). Finally, the pectin films were capable of effectively sealing linear enterotomies in a bowel simulacrum as well as an ex vivo bowel segment. We conclude that pectin is a biomaterial with physical and adhesive properties capable of facilitating anastomotic healing after intestinal surgery.

## 1. Introduction

Cancer is the most common indication for gastrointestinal surgery. In 2021, an estimated 3 million people world-wide will have a new diagnosis of gastric or colorectal cancer [1,2,3]. Although the current trend is toward multimodal therapeutic approaches including chemotherapy, biologic therapies, and radiotherapy [4,5], surgery remains the mainstay of contemporary treatment. The surgical resection of the tumor-bearing portion of the gastrointestinal tract provides important staging information as well as therapeutic value in the treatment of most gastrointestinal cancers.

Surgical resection of the tumor, however, requires the anastomosis of the two remaining ends of the bowel. This problem has been addressed with many different hand-sewn and stapled techniques of bowel anastomosis. Over the past 100 years [6,7,8], the techniques have varied widely including differences in suture material (absorbable versus non-absorbable), suture placement, single- versus double-layers and continuous versus interrupted sutures [8]. Despite the variability in the anastomotic details, a consistent feature of both hand-sewn and stapled techniques is the universal objective to achieve reinforced serosal apposition [7].

The dominant complication of gastrointestinal surgery is anastomotic failure; namely, an anastomotic leak. Anastomotic leakage has been a frequent complication of intestinal surgery for over a century [9,10,11,12]. Anastomotic leaks are generally not related to instrument or technical failures. Anastomotic leaks occur 5–10 days after construction of the surgical anastomosis [13,14], that is, at the peak of tissue remodeling. The timing of anastomotic failures suggests that leaks are a result of inadequate mechanical support of the anastomosis during the vulnerable remodeling phase of wound healing [15]. Although a variety of approaches have been tried—including fibrin glue [16], staple-line reinforcement [17,18], suture-line reinforcement [19,20], and native tissue reinforcement [21,22]—there is currently no generally accepted mitigation strategy for anastomotic leaks [23].

A major challenge for biomaterial-assisted anastomotic healing is the serosal surface layer. The serosal surface is a barrier composed of a glycocalyx or mesopolysaccharide (MPS). The MPS is 100 to 1000-fold thicker than previously demonstrated. More than 10 μm thick, the MPS represents a substantial physical barrier [24]. Most likely an adaptation to facilitate unencumbered peristaltic movements within the abdominal cavity, the serosal MPS is non-adhesive and slippery. Biomaterials used to buttress the anastomosis such as fibrin glues have failed, in part, due to limited adhesivity to the serosal surface [16].

Pectin is a plant-derived biopolymer patch potentially capable of reinforcing bowel anastomoses and limiting anastomotic leaks. Pectin is a structural heteropolysaccharide that comprises a significant portion of the primary cell walls of plants [25]. The most abundant component of pectin is homogalacturonan, a glycan of a1 → 4-linked D-galacturonic acid that can be carboxy methyl esterified [26,27]. Pectin also has the functional property of being a bio-adhesive. We have previously shown that high methoxyl pectin adheres to the surface of visceral organs including bowel lung heart and liver [28]. The adhesion between MPS and pectin appears to be the result of mutual branched-chain entanglement [29,30].

Pectin offers an intriguing potential solution to anastomotic leakage since it provides a mechanical reinforcement of the anastomosis at the critical serosal MPS interface. Pectin may also minimize the anastomotic wall tension believed to contribute to local tissue ischemia [31]. In addition, pectin is extensible likely facilitating peristaltic movement and bowel contraction [30,32].

In this report, we investigated the potential utility of pectin as an anastomotic biomaterial by evaluating the adhesivity and burst strength of pectin patches adherent to the small bowel serosa.

## 2. Methods

### 2.1. Animals

Male mice, 35 gm wild-type C57BL/6 (Jackson Laboratory, Bar Harbor, ME, USA), were anesthetized prior to euthanasia [33] and their care was consistent with guidelines of the American Association for Accreditation of Laboratory Animal Care (Bethesda, MD, USA) and approved by the Brigham and Women’s Hospital Institutional Animal Care and Use Committee. Bowel adhesion assays were performed with a porcine small bowel, which was procured by a local vendor (Research 86, Boston, MA, USA) and studied with a protocol approved by the Brigham and Women’s Hospital Institutional Animal Care and Use Committee.

### 2.2. Lectin Histochemistry

Cryostat sections were obtained from bowel specimens perfused with O.C.T. compound and snap frozen. After warming the slide to 27 °C, the sections were fixed for 10 min (2% paraformaldehyde and PBS at pH 7.43). The slides were washed with buffer (PBS, 5% sheep serum, 0.1% azide, 1 mM MgCl_2_, 1 mM CaCl_2_) and blocked with 20% sheep serum in PBS. The slides were treated with the biotinylated lectin followed by avidin-fluorescein (Southern Biotech, Birmingham, AL, USA) or avidin-fluorescein alone control. The slides were incubated for 1 h at 27 °C, washed 3 times, and mounted with DAPI-containing medium (Vector Labs, Burlingame, CA, USA). The tissue sections were imaged with a Nikon Eclipse TE2000 inverted epifluorescence microscope (Tokyo, Japan).

### 2.3. Lectins

The biotinylated lectins were obtained from commercial sources. Concanavalin ensiformis (ConA) [34] were obtained from Dako (Carpinteria, CA, USA). The lectin obtained from Vector Laboratories (Burlingame, CA, USA) was Solanum tuberosum (STA) [35].

### 2.4. Pectin

The citrus pectins used in this study were obtained from a commercial source (Cargill, Minneapolis, MN, USA) as described [32]. Briefly, the degree of methoxylation reflected the proportion of galacturonic acid residues in the methyl ester form determined. High-methoxyl pectins (HMP) were defined as those pectin polymers with a greater than 50% degree of methoxylation (mean = 70 ± 5%) [36]. The pectin films had a glycosyl residue content of 73–84% mole % galacturonic acid, 3–5% rhamnose, 19–16% galactose, and 3–5% arabinose based on gas chromatography–mass spectrometry. The pectin powder was stored in low humidity at 25 °C.

### 2.5. Pectin Dissolution in Water

The pectin powder was dissolved at 25 °C by a controlled increase in water as previously described [37]. Briefly, the complete dissolution of the pectin was obtained using a high-shear 10,000 rpm rotor-stator mixer (L5M-A, Silverson, East Longmeadow, MA, USA). The dissolved pectin was poured into standardized molds and cured to glass phase films for further studies as previously described [30].

### 2.6. Nanocellulose Fibers (NCF)

Briefly, NCF was obtained from the University of Maine (Process Development Center, Orono, ME, USA). The NCF dissolution was obtained with progressive hydration followed by high-shear 10,000 rpm rotor-stator mixer (L5M-A, Silverson). The NCF powder was dissolved at 25 °C by a controlled increase in water similar to pectin. The dissolved NCF was poured into standardized molds and cured for further studies.

### 2.7. Pressure-Sensitive Adhesive (PSA)

The PSA was a proprietary multi-purpose acrylic adhesive made available through the cooperation of the 3M Corporate Research Materials Laboratory (St. Paul, MN, USA).

### 2.8. Surgical Sealants

The various surgical sealants used in ex vivo testing include SepraFilm (Baxter, Deerfield, IL, USA), DuraSeal (Integra LifeSciences, Plainsboro, NJ, USA), Evicel (Ethicon, Somerville, NJ, USA), Surgicel (Ethicon), and Coseal (Baxter).

### 2.9. Adhesion Testing: Tensile Adhesion Strength

The pectin-serosal adhesion experiments were performed with a custom fixture designed for the TA-XT plus with 50 kg load cell (Stable Micro Systems, Godalming, Surrey, UK). The fixture was composed of a 30 mm diameter flat-ended stainless-steel cylindrical probe and a flat stainless-steel fixture surface. Very thin serosal samples (~1 mm thick and 30 mm in diameter) were harvested by sharp dissection and mounted on the fixture platform. The pectin film was, in turn, mounted to the probe surface. The cylindrical probe velocity range was 0.5 mm/s to 10 mm/s. The probe descended at the selected probe velocity for a distance of 1.5 mm above the film plane. The probe compressed the pectin and serosa at a compression force of 5 N and development time of 30 s. The probe was then withdrawn at 0.2 mm/s with continuous force and distance recordings at 500 points per second (pps).

### 2.10. Adhesion Testing: Shear Resistance

The pectin-serosal shear resistance experiments were also performed with a custom fixture designed for the TA-XT plus with 50 kg load cell (Stable Micro Systems, Godalming, Surrey, UK). The fixture was composed of separate mounts for the pectin and the serosa. The standard 30 mm diameter surface mounts were gently compressed for 30 s. The mounts were subjected to a 180° shear force at 0.5 mm/s. Constant force and distance measurements were recorded at 500 pps.

### 2.11. Transmission Electron Microscopy

The samples were stained and processed as previously described using ruthenium red [24]. Briefly, samples were stained with ruthenium red (Sigma-Aldrich, Steinheim, Germany) while in 1% glutaraldehyde, then further processed with the rOTO protocol before embedding in Epon (Serva, Heidelberg, Germany). In addition, 70 nm ultrathin sections of the tissue were analyzed using the JEOL JEM-1400 electron microscope (JEOL, Tokyo, Japan) operating at 80 kV and equipped with a 4K TemCam F416 CMOS camera (Tietz Video and Image Processing Systems GmBH, Gautig).

### 2.12. Static Pressure Simulacrum

The simulacrum used for testing the serosal sealant was a custom-designed polypropylene cylinder. The mounting chamber contained two Mediprene thermoplastic elastomer mounting gaskets (Hexpol TPE, Sandusky, OH, USA). The mounting chamber had a 0.785 cm^2^ cross-sectional area and was exposed to a 12 cm H_2_O fluid column. A stab enterotomy was performed with rapid efflux of column fluid. The enterotomy was sealed with a pectin film and exposed to a modest static pressure for more than 1 h (12 cm H_2_O).

### 2.13. Luminal Pressure Simulation

Jejunal segments 30 cm in length were obtained with an occlusive surgical clamp on one end and a luminal cannula secured with a purse-string surgical suture (0-silk, Ethicon) reinforced with a vascular tape (Ethicon). A templated 2 cm transmural enterotomy was made in the mid-portion of the jejunal segment. In experimental conditions, the enterotomy edges were manually approximated and sealed with the appropriate sealant. The lumen was gradually pressurized to 100 cm H_2_O. The pressure plateau was maintained for 10 min when possible. The plateau time was measured for each condition. A minimum of 3 replicates were performed for each simulation. Due to anatomic and ischemic variability of luminal pressure testing, a representative result is shown.

### 2.14. Statistical Analysis

The statistical analysis was based on measurements in at least three different samples. The unpaired Student’s *t* test for samples of unequal variances was used to calculate statistical significance. The data were expressed as mean ± one standard deviation. The significance level for the sample distribution was defined as *p* < 0.01. A minimum of triplicate samples were analyzed per condition in adhesion assays. Anatomic and mural variability, as well as variability in ischemic times, precluded reliable replicates in the luminal pressure testing.

## 3. Results

### 3.1. Pectin and MPS Adherends

Pectin films, prepared at a range of water contents, were translucent and flexible (Figure 1A). To assess the optimal water content, the pectin films were tested for burst strength and extensibility using a uniaxial load applied at constant velocity normal to the plane of the film until fracture (Figure 1B). The burst strength of the films was relatively independent of water content (Figure 1C), but extensibility was highly sensitive to water content (Figure 1D). The pectin films used in this study were prepared at an intermediate water content (Figure 1C,D; gray box).

The hydrated MPS on the surface of visceral organs is responsible for the glistening appearance and slippery surface of the gastrointestinal tract. The distinctive carbohydrate-rich MPS is readily detected by lectins which bind to specific mono- and oligosaccharides. Using fluorescein-labeled lectins, we have demonstrated comparable lectin-binding patterns on the MPS of humans, sheep, swine, and mice intestinal organs (not shown) (Figure 2). On the ultrastructural scale, transmission electron microscopy (TEM) and specific stains such as ruthenium red have been used to depict the glycocalyx on various organ surfaces, including the bowl lumen, the capillary endothelium, and the epithelium of the nose and bronchi [38]. Here, we were able to visualize the characteristic densely stained aggregates in a periodic distribution on the mesothelial cell surface, on the surface of serosal microvilli, and on secretory pits and vesicles (Figure 2C,D). The findings are in line with pervious descriptions of cell surface polysaccharide morphology [39].

### 3.2. Tensile Adhesion Strength

The primary requirement for bowel anastomotic reinforcement is pectin film adhesivity to the MPS. To test for pectin adhesion to the bowel wall, we screened serosal samples from mice, pig, and human small bowel. Although each species demonstrated subjectively comparable binding, porcine samples were chosen due to slaughterhouse availability, specimen size, and replicate opportunities (Figure 3A,B). The pectin-mounted probe was lowered at constant velocity until a trigger compression force was achieved (5N). The compression force was maintained for a 30 s development time followed by probe withdrawal at 0.5 mm/s. The striking observation was the elasticity of the subserosal tissue. Probe withdrawal resulted in the stretching of the thin subserosal tissue with a resulting protracted adhesion curve (Figure 3C). In most assays, avulsion of the subserosal tissue occurred before any debonding was observed between the pectin and the MPS. Despite these quantitative limitations, pectin was significantly more adhesive to porcine serosa than NCF or PSA (*p* < 0.001) (Figure 3D).

### 3.3. Shear Resistance

A secondary requirement for bowel anastomotic reinforcement is pectin film resistance to shear forces produced by bowel peristalsis. To assess shear resistance to forces parallel to the plane of adhesion, we mounted small bowel serosa and pectin films to a custom fixture (Figure 4A,B). The pectin and serosa were gently compressed for 30 s prior to distraction of the surfaces at 0.5 mm/s, that is, a 180° applied shear force. Similar to the observations during tensile strength testing, there was significant elasticity of the porcine subserosal tissue resulting in a protracted adhesion curve (Figure 4C). The work of adhesion, reflecting the area under the adhesion curve, was highly significant when compared to NCF and PSA (*p* < 0.001) (Figure 4D).

### 3.4. Luminal Pressure Simulation

To assess the capability of pectin to seal enterotomies, we used two different ex vivo assays. The static pressure simulacrum assay permitted a convenient comparison of pectin adhesion to serosal segments from different animals and small bowel segments. Despite variability in ischemic times from 1 to 6 h, there were no sealant failures (Figure 5A). The second assay pressurized a small bowel segment after sealing a 2 cm enterotomy with pectin film. The bowel segment was sealed with pectin or other purported sealants prior to pressurization to 100 cm H_2_O for 10 min. The assay was limited to 10 min, due to pressure-induced structural changes noted in the bowel wall. Commercial surgical adhesives, such as EviCel (Ethicon) and DuraSeal (Integra Life Sciences, Plainsboro, NJ, USA), failed within seconds. SepraFilm (Baxter), marketed as an adhesion barrier, was unexpectedly effective in 10 min (Figure 5B). Notably, SepraFilm is a linear chain polysaccharide polymer.

## 4. Discussion

In this report, we studied the serosal adhesivity of the plant-derived heteroplysaccharide pectin. As a biopolymer, pectin has the adhesivity and cohesion necessary for anastomotic reinforcement. Pectin also has the extensibility to accommodate intestinal movements. Although quantitative measurements of pectin-serosal adhesion were compromised by the elasticity of the subserosal tissue, our studies demonstrated pectin’s substantial tensile adhesion strength, shear resistance, and pressure tolerance. Moreover, pectin was significantly more adhesive than either NCF films or PSA (*p* < 0.001). We conclude that pectin is a biomaterial with physical and adhesive properties capable of facilitating anastomotic healing.

Anastomotic healing is challenged by the mechanic forces produced by the bowel wall. Coordinated muscular contraction of the circular and longitudinal muscles not only moves luminal contents along the digestive tract, but also generates forces disruptive to anastomotic healing. The longitudinal and circular muscles generate an average luminal pressure of 20 mmHg [40]. Contractile waves occur at a frequency of 7–20/min [41,42] and are propagated at 3 cm/s [43]. These luminal pressure waves are aggravated by postoperative healing [44] and underlying disease states [45,46,47]. Moreover, luminal contents are minimally compressible ensuring that the mucosal and submucosal layers are squeezed between contracting muscles [48].

Surgeons have long recognized that atraumatic serosal apposition [31], imbrication [49], invagination [50], and buttressing [51] facilitates anastomotic healing, however, only recently has the anatomy of the mesothelial surface been clarified [24]. The serosal surface of the bowel wall is covered with a serosal MPS. We have recently demonstrated that the MPS is 1000-times thicker than previously assumed—over 10 μm in thickness [24]. Moreover, high resolution imaging indicates that the serosal MPS demonstrates a branched-chain structure that mirrors the chain structure of pectin. Reflecting the polymer axiom that like-entangles-with-like, pectin appears to bind to the serosal MPS by a process of branched-chain entanglement [30]. This branched-chain entanglement, analogous to the hook-and-loop mechanism of commercial Velcro, produces the significant adhesion tensile strength and shear resistance of pectin-MPS adhesion.

There are several potential benefits of applying a pectin patch to the surface of a freshly constructed bowel anastomosis (Figure 6). The primary benefit is mechanical reinforcement that ensures serosal apposition with distributed tissue strain. A broad reinforcement surface area minimizes tension at the edges of the anastomosis and maximizes tissue perfusion. In contrast, standard surgical sutures and staples are associated with significant localized strain patterns at the edges of the anastomosis where blood perfusion is tenuous. A second potential benefit is a physical barrier function provided by the pectin patch. The blending of the pectin and endogenous MPS re-establishes the pre-existing MPS barrier that functions to maintain tissue hydration, seal microperforations, and minimize intra-abdominal adhesions. Finally, the interaction between mesothelial cells and the MPS suggests a potential scaffold function for the MPS. By bridging the serosal defect, pectin may facilitate tissue repair across serosal defects.

The limitations of this study reflect the difficulty of simulating the complex physiological forces borne by the bowel anastomosis. We identified three desirable features for anastomotic reinforcement: Tensile adhesion strength, shear resistance, and luminal pressure resistance. Tensile adhesion strength reflects basic adhesivity necessary to establish the wound interface. Shear strength provides insights into sealant resistance to complex peristaltic forces. Resistance to luminal pressurization characterizes the sealant performance in a simulation of clinical bowel distension and barrier function. Attempts to isolate these functions were compromised by the significant elasticity of the subserosal tissue. Serosal (or adventitial) anatomy of the bowel wall has traditionally been described as an outer mesothelial (or serosal) layer and an inner subserosal layer comprised of blood vessels, nerves, lymphatics, and elastin-containing connective tissue [52]. The extensibility of the subserosa will be an ongoing challenge in future studies of serosal bioadhesives.

In summary, pectin films demonstrated significant tensile strength adhesion and shear resistance. Pectin films also demonstrated an effective seal of pressurized small bowel segments. We conclude that pectin is a biomaterial with physical and adhesive properties capable of facilitating anastomotic healing after intestinal surgery.

## Figures and Tables

**Figure 1 polymers-13-02811-f001:**
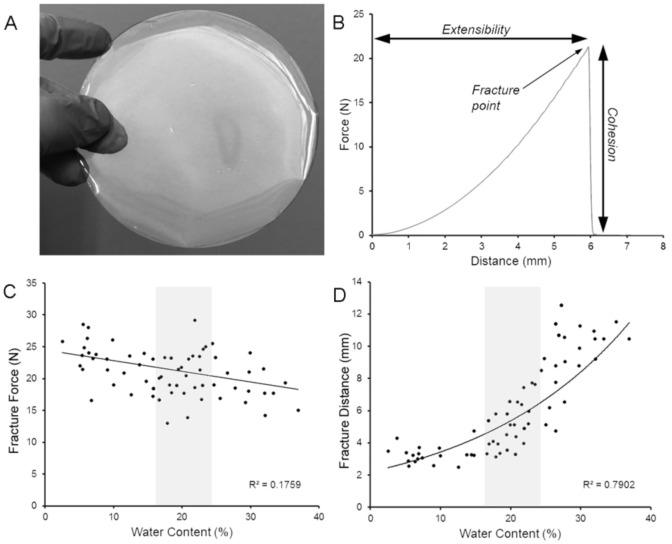
Physical properties of pectin films used for bowel adhesion. (**A**) The pectin films were poured into a custom mold prior to curing. The films were translucent and flexible. (**B**) As previously described [32], burst testing was performed with a stainless-steel spherical probe that descended at a velocity of 0.5 mm/s until film rupture (fracture point). Continuous force and distance measurements were recorded at 500 points per second (pps). Burst testing provided a measure of pectin cohesivity (**C**) and extensibility (**D**) at various water concentrations. The pectin films used in this study were selected with intermediate physical properties (gray rectangle).

**Figure 2 polymers-13-02811-f002:**
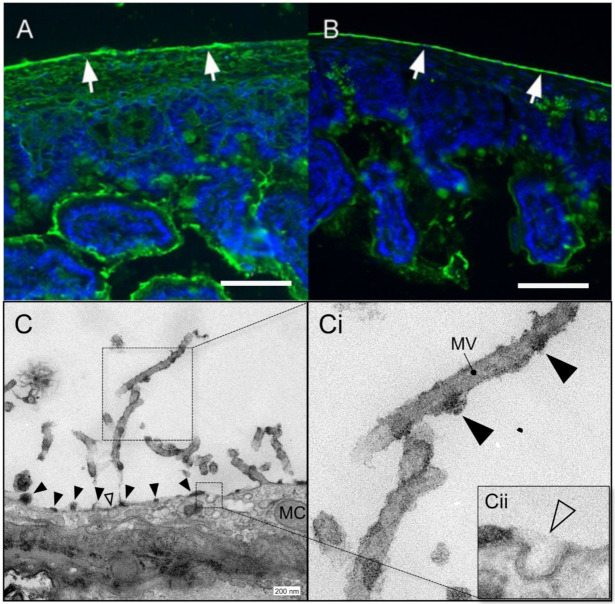
Mesopolysaccharide (MPS) of murine small bowel MPS. The glucose/mannose group lectin Concanavalin ensiformis (ConA) [34] (**A**) and the N-acetylglucosamine group lectin Solanum tuberosum (STA) [35] (**B**) were used to demonstrate carbohydrates on the serosal surface (white arrows; bar = 50 µm). (**C**) Transmission electron microscopy (TEM) of the serosa demonstrating mesothelial cells (mc) and microvilli (mv). Higher resolution image (**Ci**) demonstrating findings consistent with MPS attached to the microvilli. Membrane vesicles and pits potentially involved in MPS hydration and maintenance are seen beneath the cell surface (**Cii**, inset).

**Figure 3 polymers-13-02811-f003:**
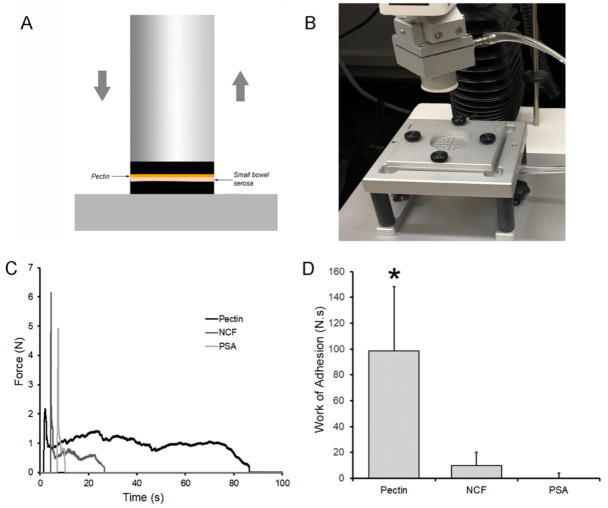
Tensile strength adhesion testing ex vivo. (**A**,**B**) A custom fixture with a pectin mounted 30 mm flat surface probe was used to engage the 1 mm thick serosal specimen mounted on the fixture platform. After 5N compression for 30 s development time, the probe was withdrawn with 500 pps force and distance recordings. (**C**) Due to the extensibility of the subserosal tissue, the pectin-serosal adhesion resulted in a protracted adhesion curve (black line). In contrast, the nanocellulose fiber (NCF) film (dark gray) and pressure sensitive adhesive (PSA) (light gray) debonded shortly after the onset of withdrawal. (**D**) Replicate samples demonstrated that the work of adhesion, representing the area under the force-distance curve, was significantly greater in the pectin samples compared to NCF and PSA (*, *p* < 0.001).

**Figure 4 polymers-13-02811-f004:**
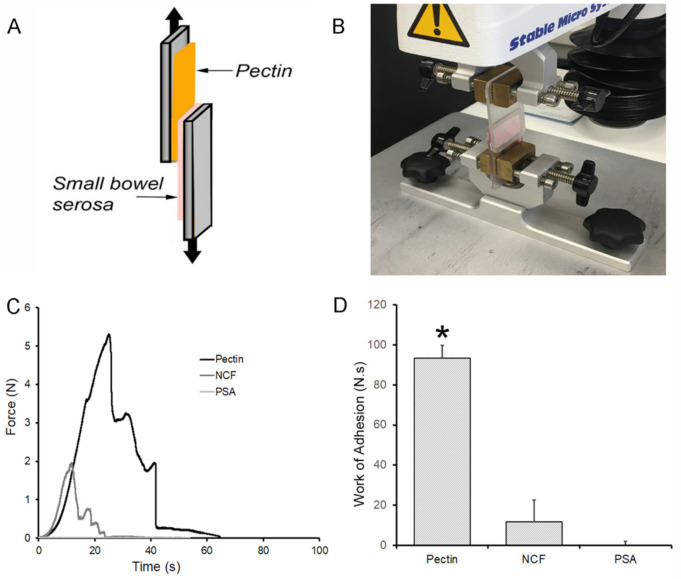
Shear strength adhesion testing ex vivo. (**A**,**B**) A custom fixture with a pectin films and 1 mm thick porcine serosa mounted on opposing surfaces. After gentle compression (approximating 5N force) for 30 s development time, the surfaces were withdrawn at 180° at 0.5 mm/s with 500 pps force and distance recordings. (**C**) Due to the extensibility of the subserosal tissue, the pectin samples resulted in a protracted adhesion curve (black line) in contrast to the nanocellulose fiber (NCF) film (dark gray) and pressure sensitive adhesive (PSA) (light gray). (**D**) Replicate samples demonstrated that the work of adhesion, representing the area under the force-distance curve, was significantly greater in the pectin samples compared to NCF and PSA (*, *p* < 0.001).

**Figure 5 polymers-13-02811-f005:**
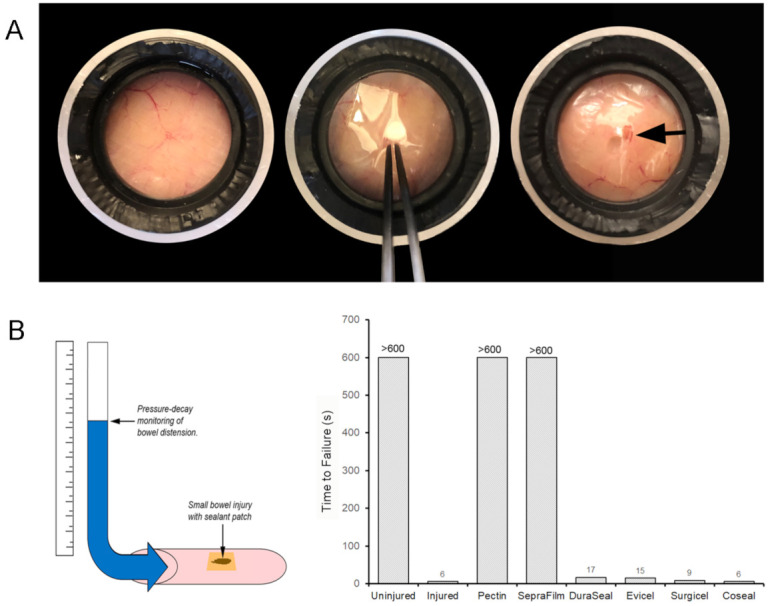
Simulation of surgical injury ex vivo. (**A**) Uninjured serosa (*i*) was perforated with a no. 15 scalpel blade (Swann-Morton, Sheffield, England) stab enterotomy followed by application of the pectin patch (*ii*). The patch repair was exposed to 12 cm H_2_O static pressure for 1 h with no detectable leak (*iii*). (**B**) A 3 cm linear enterotomy was created in a 30 cm porcine small bowel segment. The enterotomy was sealed with a pectin patch and various surgical sealants including SepraFilm (Baxter, Deerfield, IL, USA), DuraSeal (Integra Life Sciences, Plainsboro, NJ, USA), Evicel (Ethicon, Somerville, NJ, USA), Surgicel (Ethicon), and Coseal (Baxter). The intraluminal pressure of the bowel segment was increased with a 10 s ramp to a 100 cm H_2_O plateau pressure for 10 min. Most sealants failed to reach plateau pressures; the label denotes the time (seconds) to pressure reversal in a representative segment as shown.

**Figure 6 polymers-13-02811-f006:**
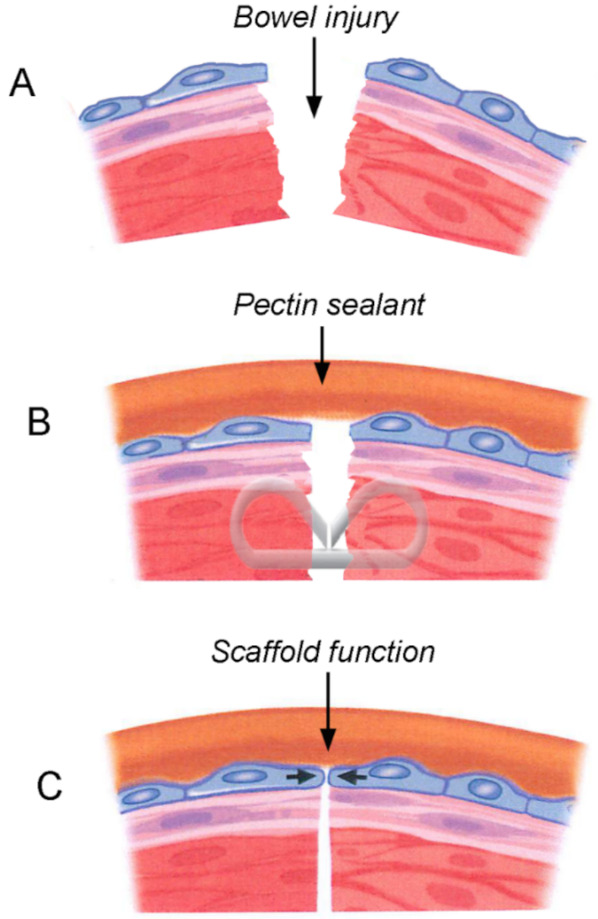
Schematic of the potential function of the pectin biopolymer after bowel injury (**A**). The pectin may function as a sealant with mechanical and barrier functions (**B**). Pectin may also function as a wound healing scaffold (**C**).

## Data Availability

Not applicable.

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
