# Peer review of "Biomaterial-Assisted Anastomotic Healing: Serosal Adhesion of Pectin Films"

_polymers, 2021, doi:10.3390/polym13162811_

Round 1

Reviewer 1 Report

I have the following suggestions for authors:

  1. Figure 3D and 4D: missing statistically significant differences.
  2. Figure 5A, 5B: There is no discussion in the manuscript describing these figures. There should be added in Materials and Methods how many repetitions of the simulation of surgical injury was done.
  3. In some parts of Results section, the authors are describing how they perform the analysis, but this should be only in part Materials and Methods, please correct.
  4. The conclusion is missing.

Reviewer 2 Report

The manuscript is relativelly well written. There are present only a few minor technical points which decrease its overall quality.

  1. line 77: According to the guide for authors, there has to be mentioned city and country of the origin for the inverted epifluorescence microscope.
  2. Lines 106, 133 as well as 135: countries of the origin have to be mentioned.
  3. Fig 2. caption: "50 um" has to be rewritten to "50 μm".

Thus, the above mentioned manuscript will be ready to be published after minor revision.

Round 2

Reviewer 1 Report

The manuscript can be accepted.